# Effects of the Resilience of Nurses in Long-Term Care Hospitals during on Job Stress COVID-19 Pandemic: Mediating Effects of Nursing Professionalism

**DOI:** 10.3390/ijerph181910327

**Published:** 2021-09-30

**Authors:** Bom-Mi Park, Jiyeon Jung

**Affiliations:** 1Department of Nursing, Konkuk University, Chungju-si 27478, Korea; spring0317@kku.ac.kr; 2Department of Nursing, Korea National Open University, Seoul 03087, Korea

**Keywords:** COVID-19, occupational stress, professionalism, long-term care

## Abstract

Purpose: To investigate nursing professionalism as a mediating factor in the relationship between resilience and job stress levels for nurses working in long-term care hospitals during the COVID-19 pandemic. Methods: A cross-sectional survey was conducted from January to March 2021 in seven long-term care hospitals in the Seoul metropolitan area to measure resilience, nursing professionalism, and job stress among nurses. Simple and multiple regression analyses along with the Sobel test were performed to verify the mediating effect of nursing professionalism. Results: Data from 200 nurses were included in the final analysis. Results showed that individual and occupational characteristics could lead to differences in nurses’ resilience, job stress levels, and nursing professionalism. Nursing professionalism had a significant mediating effect on the relationship between resilience and job stress levels. The effect of resilience on job stress levels was significant (β = −0.16, *p* = 0.024). After controlling for nursing professionalism, the effect declined and was not statistically significant (β = −0.09, *p* = 0.251). Conclusion: There is a need to increase individual resilience and nursing professionalism through intervention programs and policy proposals to manage job stress among long-term care hospital nurses during the COVID-19 pandemic.

## 1. Introduction

COVID-19, a severe acute infectious respiratory disease, has been rapidly spreading worldwide [1]. During the pandemic, the number of COVID-19 patients in long-term care (LTC) facilities for the elderly, including LTC hospitals, has rapidly increased in Korea [2]. This situation indicates the lack of infection monitoring systems [3] and reveals insufficient human resources for infection control in LTC facilities [2,4].

In this context, nurses in LTC facilities have reported fear of infection, the burden of duty, and emotional exhaustion [5]. It may be more burdensome for these nurses to work without clear practical guidelines relevant to the pandemic [3], and they face more significant difficulties as they must be in daily contact with their patients [6]. There is an urgent need to develop strategies for overcoming excessive workload, fear, and stress in pandemic [1].

Hence, this study focused on nurses’ resilience to overcome these challenging conditions. The Resilience Development Model of Grafton et al. [7] suggests that nurses’ resilience can be developed through cognitive transformation, personal growth, educational and environmental support—the so-called “process of increasing resilience”. Furthermore, nurses’ job stress can adversely impact this development process. In short, it is hypothesized that nurses’ resilience can be enhanced through professional growth and support for mitigating the impact of job stress. Based on the model by Grafton et al. [7], this study conceptualized a positive cognitive factor called nursing professionalism as a mediating variable in the relationship between nurses’ resilience and job stress.

Nursing professionalism comprises nursing values and a sense of mission [8]. Nursing professionalism may determine the quality of nursing care through increased job satisfaction [9], job performance [10], and commitment [11]. According to the literature, nursing professionalism has played an important role during the pandemic to empower others and provide qualitative care [12]. It is primarily required for appropriate communication and patient satisfaction amid the pandemic [13]. Studies indicate that nurses’ professionalism is an essential determinant of their work performance [14], and it also affects nursing intention [15]. Most importantly, nursing professionalism can build resilience, which has been shown to counter distress during the COVID-19 pandemic [16]. 

In short, resilience is a dynamic, innate process to empower and recover from job stress [7], and it can be influenced by personal and work-related factors [17,18]. Moreover, resilience can play an essential role in overcoming and adapting to difficulties or stressful situations and establishing a professional nursing identity [19,20]. A high level of resilience increases nursing performance [21], professional identity [20], nursing professionalism [22], and lowers job stress [21] and burnout among nurses [17]. In other words, nurses’ ability to use internal resources can be very effective for managing job stress [23]. 

Due to poor conditions in LTC hospitals during the COVID-19 pandemic [2], levels of job stress experienced by LTC nurses are considered relatively high. There have been studies on nurses’ experiences in COVID-19 designated hospitals [1] and their stressful situations during the pandemic [24]. However, few studies have been carried out on nurses working with elderly patients, who are the most susceptible to infection. 

Based on this background, we hypothesized that nurses’ resilience will affect the reduction in job stress, and nursing professionalism can be a parameter in this correlation. Accordingly, we attempted to verify the influence of nursing professionalism as a mediating factor in the relationship between resilience and job stress among LTC hospital nurses in the context of the COVID-19 pandemic. 

## 2. Materials and Methods

### 2.1. Study Design

This study used a cross-sectional descriptive design to identify the effects of LTC hospital nurses’ resilience related to job stress and the mediating effects of nursing professionalism in the context of the COVID-19 pandemic.

### 2.2. Setting and Sample

The participants in this study included nurses working in an LTC hospital for over 3 months during the COVID-19 outbreak. Based on previous study, working over 3 months can be considered a sufficient period for nurses to independently perform their duty and adapt to the workplace [25]. In this study, there were no other inclusion criteria except for working periods. We recruited nurses from seven LTC hospitals with over 150 beds in the Seoul metropolitan area. The sample size was estimated at a power of 0.95, a significance level of 0.05, a medium effect size of 0.15, and through 13 predictor variables using G*Power 3.1.9.2 for the multiple regression analysis. The required minimum number was 189 participants. 

### 2.3. Measurements

This study measured general characteristics, including socio-demographic and occupational characteristics, among the participants. Nurses’ resilience, nursing professionalism, and job stress were treated as critical variables in this study.

Socio-demographic characteristics included age, gender, marital status, education level, religion, and cohabitant(s). Occupational characteristics comprised job position (staff, charge, head, and chief nursing officer), overall working years as a nurse, and working years in an LTC hospital. Additionally, having experience(s) of COVID-19 education in the organization (yes or no), satisfaction with the education contents, and nursing readiness during the COVID-19 pandemic using a 4-point Likert scales were measured.

This study investigated the critical variables using valid instruments. We obtained approvals for the utilization of all the instruments from the authors.

Resilience was measured using the Korean version of the Connor–Davidson Resilience Scale (K-CD-RISC) developed by Connor and Davidson [26] and translated by Baek [27]. This instrument comprises 25 questions across the following subdomains: hardiness, coping, adaptability, meaningfulness, optimism, regulation of emotion and cognition, and self-efficacy. Responses are given on a 4-point Likert scale with scores ranging from 0 (“not at all”) to 4 points (“very much”). The total score for the instrument ranges from 0 to 100; the higher the score, the higher the resilience level. The Cronbach’s α was 0.89 in Baek’s study [27] and 0.93 in the current study.

Nursing professionalism was measured using an instrument developed by Yeun et al. [28]. This instrument consists of 29 questions across five subdomains: social perception, self-concept of the profession, nursing expertise, nursing practice role, and nursing independence. Responses are given on a 5-point Likert scale with scoring ranging from 1 (“not at all”) to 5 points (“very much”); the higher the score, the higher the nursing professionalism. The total score for this tool ranges from 29 to 145. The Cronbach’s α was 0.92 in Yeun et al.’s study [28] and 0.91 in the current study. 

Job stress was measured using an instrument for nursing job stress developed by Kim and Gu [29] and revised by Choi [30]. This instrument consists of 32 questions across seven subdomains: workload, lack of expertise and skills, inappropriate compensation, psychological burden, role conflict, physical environment, and work and interpersonal relations. Responses are given on a 5-point Likert scale, with scores ranging from 1 (“I do not feel at all”) to 5 (“I feel very badly”). The total score ranges from 32 to 160; the higher the score, the higher the level of job stress. The Cronbach’s ⍺ was 0.95 in Choi’s study [30] and 0.95 in the current study.

### 2.4. Data Collection and Procedure

Data were collected from 29 January to 2 March 2021. The self-report questionnaire survey was conducted with the cooperation of the nursing management departments in designated hospitals after providing necessary explanations regarding the study. The survey was administered randomly to nurses interested in this study and who expressed their wish to participate when recruited. Moreover, it was made available online via Google Forms, which was considered appropriate in the COVID-19 context. 

The explanations about the study were distributed across the nursing departments, and additional explanations were provided via phone call as required. After that, a survey was conducted among nurses who voluntarily agreed. First, the researchers explained the details of participating in the study. Further, the participants were informed that there would be no disadvantages for not participating or withdrawing. It was also confirmed that the survey results would not be used for any other purpose than the research. It took about 20 to 30 minutes to fill out the questionnaire.

### 2.5. Ethical Considerations

Ethical considerations for the overall study were reviewed and approved by the Institutional Review Board (IRB No. 7001355-202101-HR-415). This study was also registered with the Clinical Research Information Service of the Republic of Korea (PRE20210204-004). Before data collection, the researcher communicated with each hospital’s nursing department to explain the purpose of the study and the contents of the questionnaire and promised cooperation. Participants could contact the research director if they had any inquiries about the questionnaire. We obtained consent from the participants after notifying them that their responses would be kept confidential. Upon completion of the survey, each participant received a small fee.

### 2.6. Data Analysis 

Descriptive statistics with means, frequencies, and standard deviations (SD) were used to quantify the participants’ characteristics and critical study variables. We performed an exploratory analysis with a degree of scattering, skewness, and kurtosis for statistical analysis of the continuous variables included in this study. In addition, the normal distribution of continuous variables was confirmed using the Shapiro–Wilk test. According to the participants’ general characteristics, the differences in dependent variables were analyzed using the Student’s *t*-test, Pearson’s correlation, and one-way analysis of variance with post hoc analysis using Scheffé and Dunnett T3 methods. Additionally, for variables that did not satisfy normality in the Shapiro–Wilk test, nonparametric Mann–Whitney test and Kruskal–Wallis test were used. The Cronbach’s α was measured to ensure instrument reliability.

Pearson’s correlation coefficient was used to analyze correlations between resilience, professionalism, and job stress levels. To verify the mediating effect of nursing professionalism in the relationship between job stress levels and resilience, simple and multiple regression analyses were performed based on the three-step procedure proposed by Baron and Kenny [31]. In addition, the Sobel test was conducted to confirm the significance of the mediating effect.

Data were analyzed using SPSS Statistics software version 25.0 (SPSS, IBM Corp, Chicago, IL, USA). The level of statistical significance was set at *p* < 0.05.

## 3. Results

### 3.1. Participants’ General Characteristics and Differences in Study Variables by Characteristics

Among the 208 recruited participants, 8 were excluded from the final analysis due to withdrawal and non-response to the questionnaire. Accordingly, 200 were included in the final analysis.

The analysis revealed that the participants were mostly women (99.5%), with an average age of 44.8 years, with approximately 70% of them aged 40 years or older. About 62% of the participants were married, and more than half reported having a religion (58.0%). Of the participants, 61.5% had bachelor’s degrees or higher, and 81.5% lived with cohabitants. In terms of occupational characteristics, staff nurses comprised the majority (77.5%), followed by head nurses (12.0%). The average working years were 13.4, and the average working years in an LTC hospital were 6.5. About 92.0% of the participants had received COVID-19-related education in their hospital and rated their satisfaction with the course as 3.5 out of 4 points. The participants felt that they had a relatively high level of COVID-19 nursing readiness, approximately 3.3 points. 

Different age groups showed significant differences in resilience (F = 5.21, *p* = 0.006); those under 40 years old had significantly less resilience than older age groups in the post hoc analysis. There were also significant differences in resilience (F = 2.45, *p* = 0.015) and nursing professionalism (F = 2.22, *p* = 0.027) by participants’ marital status. Regarding educational level, significant differences in resilience (F = 4.13, *p* = 0.018) were found for those who attended over graduate school. In terms of occupational characteristics, there was a statistically significant difference in the resilience mean between job positions (F = 3.09, *p* = 0.028), but there was no significant difference in the post hoc analysis using the Dunnett T3 method. Nurses with more than 10 years of nursing experience (F = 5.19, *p* = 0.006) had significantly higher nursing professionalism than those with 5 to 10 years of experience. The score for nursing professionalism was significantly higher for those who had received COVID-19 education (F = 2.41, *p* = 0.017) and were satisfied with it (F = 2.50, *p* = 0.013). COVID-19 nursing readiness also affected resilience (F = 2.63, *p* = 0.009), professionalism (F = 4.26, *p* < 0.001), and job stress levels (F = −2.12, *p* = 0.036) (Table 1).

### 3.2. Descriptive Statistics of the Study Variables 

The level of resilience, nursing professionalism, and job stress are summarized in Table 2. The average scores were as follows: resilience (62.2 ± 12.2), professionalism (97.8 ± 13.1), and job stress (114.8 ± 19.5).

### 3.3. Correlation between the Study Variables

Table 3 shows the results of the correlation between resilience, nursing professionalism, and job stress levels. Resilience was positively correlated with professionalism (r = 0.42, *p* < 0.001) and negatively correlated with job stress levels (r = −0.16, *p* = 0.024). In addition, professionalism was negatively correlated with job stress levels (r = −0.21, *p* = 0.003).

### 3.4. Mediating Effects of Professionalism on the Relationship between Resilience and Job Stress Levels

Table 4 presents the regression analysis results conducted to verify the mediating effects of nursing professionalism on the relationship between resilience and job stress levels. In the first step of the analysis, resilience was treated as an independent variable, and nursing professionalism was treated as a mediator variable. A significant effect was found between resilience and nursing professionalism (β = 0.42, *p* < 0.001). In the second step, the results of the regression with resilience (independent variable) and job stress levels (dependent variable) were statistically significant (β = −0.16, *p* = 0.024). In the third step, resilience (independent variable) and nursing professionalism (mediator variable) were simultaneously introduced into the regression model to predict job stress levels (dependent variable). The analysis revealed that nursing professionalism had a significant effect on job stress levels (β = −0.17, *p* = 0.026). However, the absolute value of the regression coefficient, the effect of resilience on job stress levels, declined from –0.16 to –0.09 and was not statistically significant (β = −0.09, *p* = 0.251). Therefore, nursing professionalism can be interpreted as completely mediating the relationship between resilience and job stress levels, as seen in Figure 1. Additionally, a Sobel test was performed to verify the significance of the mediating effects of nursing professionalism; the mediating effects on the relationship between resilience and job stress levels were found to be significant (Z = −2.12, *p* < 0.05).

## 4. Discussion

This study examined the relationship between resilience, nursing professionalism, and job stress in nurses working in LTC hospitals during the COVID-19 pandemic. The study examined the mediating effect of nursing professionalism on the relationship between resilience and job stress levels. 

Most inpatients in LTC hospitals are elderly with underlying chronic diseases, identified as risk factors for COVID-19 infection [32]. COVID-19 outbreaks have been reported in group facilities for the elderly, such as nursing homes and LTC hospitals [4]. However, LTC hospitals do not prioritize infection management with relevant policies and systems [33]. Moreover, LTC hospital nurses complain of high stress due to heavy workload and increased responsibility arising from workforce shortages and the physical and mental burden of nursing care to elderly patients [5,34]. Nevertheless, nurses have been coping with stress in crises through psychological adjustment, active learning, and communication [35]. The resilience of nurses is described as an essential intrinsic resource in adopting these strategies [7]. 

According to the Resilience Development Model by Grafton et al. [7], resilience can draw nurses to adapt positively to stressful situations by transforming positively to restore and strengthen the well-being of the self and reduce their vulnerability. High levels of resilience are essential for nurses to manage workplace stressors in their profession [36]. Such resilience can be developed by professionalism, as professional resilience is supported by organizational, professional, and personal strategies [7]. Moreover, nurses’ resilience can be increased in different situations and environments and be carried through to the nursing environment [37]. Since resilience can be acquired through training and education, it is necessary to research active interventions, provide psychological resources, and establish a positive work environment [17,22]. 

Additionally, nursing professionalism is an essential element that nurses must possess [25] and can increase nurses’ clinical performance and reduce stress in practice [38]. Research has shown that nursing professionalism can affect the performance of nurses, highlighting its role as a factor that affects coping with unusual situations such as the spread of infectious diseases [15]. Nurses’ professionalism may be affected by the unusual context of the COVID-19 pandemic, and the urgent situation may also positively affect the nurse’s professionalism [39]. As a motivational force, professionalism can help rebuild resilience in traumatic situations such as the COVID-19 pandemic [16].

Meanwhile, during the COVID-19 crisis, nurses have faced various challenges, including difficulties in their usual nursing practice, unpredictable daily life changes, psychological burden, and risk of infection [40]. During the prolonged pandemic, various mental health threats such as strain, fear, anxiety, and stress have reached severely high in frontline nurses [41]. Moreover, heavy job stress can adversely affect nursing work performance [21]. According to our study results, participants’ job stress levels were significantly lower when their COVID-19 nursing readiness was high. It has been found that organizational culture and situational factors can affect nurses’ job stress levels [42]. The current study supports a previous study that suggests the necessity of appropriate organizational support during the pandemic [43].

In this study, nursing professionalism positively mediates nurses’ resilience, affecting job stress levels. Resilience is an essential resource for dealing with job stress [44,45], and people with high resilience levels can recover successfully even under stressful events [43]. In addition, resilience can be rebuilt by fostering professionalism [16]. A previous systematic review concluded that the nurses’ resilience and coping factors, including personal views or attitudes as professionals (also known as professional identity), can help in effectively adapting to workplace stressors [46]. Similarly, Lu et al. [47] developed a path model, reporting that professional commitment is an antecedent of nurses’ job satisfaction and work stress. The current study showed that nursing professionalism had a positive mediating effect on resilience and a negative effect on job stress levels. Nurses’ professionalism affects their resilience and is also a helpful resource in coping with various stressful situations [16]. In other words, by improving nurses’ resilience and professionalism, they can manage job stress more effectively, which will eventually become the basis for providing high-quality nursing care. Developing programs for job stress management for nurses [1,48] and defining appropriate nursing care in the context of the COVID-19 pandemic [49] are imperative. Further research, including in LTC hospitals and facilities with vulnerable population groups, is needed. 

This study analyzed job stress experienced by LTC hospital nurses during the COVID-19 pandemic and its association with personal resilience and professionalism. This study is significant because it provides evidence on LTC hospital nurses’ psychological and occupational factors during the COVID-19 pandemic, significantly impacting daily life and healthcare services worldwide. To our knowledge, this is the first study to focus on the psychological factors of LTC nurses in this context. Despite this significance, a few limitations have been identified. 

First, the study was limited to seven hospitals in the Seoul area for convenience sampling. Using the convenience sampling method resulted in difficulties in recruiting survey respondents during the pandemic. There may also be limitations in generalizing the research results as the target group was limited. Second, as this study used a cross-sectional and descriptive design, there may be limitations in explaining the strong causal relationship between variables. Third, this study used a self-report questionnaire to obtain data. As respondents’ subjectivity may be reflected in the data, the research results may need to be interpreted in that light. Moreover, when the investigation was conducted in early 2021, infection numbers in LTC hospitals were high, and nurses experienced undue psychological burdens. Therefore, it is necessary to interpret the research results by carefully considering these circumstances. 

This empirical study of LTC hospital nurses has pointed out the need for more diverse research methods on related topics. Continuous research and implementation of intervention programs and policy proposals are essential for managing job stress levels, resilience, and professionalism of nurses suffering amid the COVID-19 pandemic. 

## 5. Conclusions

The level of job stress experienced by nurses in LTC hospitals during the COVID-19 pandemic is significant. This study confirmed that resilience as an individual control mechanism reduces job stress levels. Additionally, as a motivational force, nursing professionalism reduces job stress levels as a complete mediator. Therefore, improving individual resilience and nursing professionalism is essential for managing LTC hospital nurses’ job stress during the pandemic.

## Figures and Tables

**Figure 1 ijerph-18-10327-f001:**
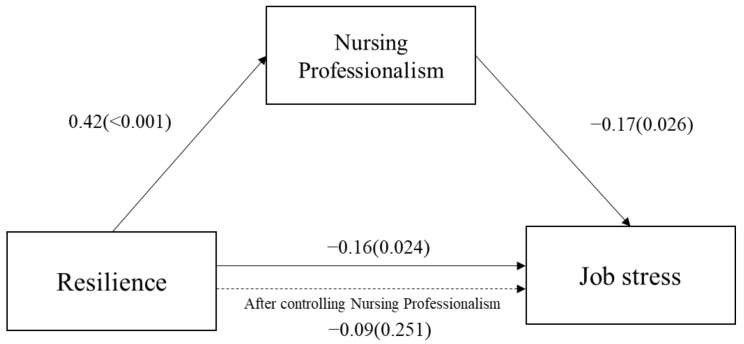
The mediating relationship between resilience and job stress levels.

**Table 1 ijerph-18-10327-t001:** General characteristics of the study participants, and differences in resilience, nursing professionalism, and job stress by characteristics (*n* = 200).

Characteristics		N (%) or Mean ± SD	Resilience		Professionalism		Job Stress	
			Mean ± SD	t/F (*p*)	Mean ± SD	t/F (*p*)	Mean ± SD	t/F (p)
Gender	Female	199 (99.5)	62.27 ± 12.22	1.00 (0.318)	97.6 ± 13.0	–2.10 (0.037)	114.9 ± 19.5	0.76 (0.447)
	Male	1 (0.5)	50.0 ± 0.0		125.0 ± 0.0		100.0 ± 0.0	
Age ^†,‡^		44.8 ± 10.8						
	< 40 ^a^	61 (30.5)	58.1 ± 11.1	5.21 (0.006) a < b, c ^‡^	94.2 ± 13.8	4.06 (0.019) a < c ^‡^	113.3 ± 18.0	1.74 (0.179)
	40–49 ^b^	61 (30.5)	63.9 ± 12.3		97.8 ± 13.7		118.7 ± 20.0	
	≥50 ^c^	78 (39.0)	64.1 ± 12.3		100.5 ± 11.6		113.0 ± 20.0	
Marriage	Yes	124 (62.0)	63.8 ± 12.5	2.45 (0.015)	99.4 ± 12.5	2.22 (0.027)	116.6 ± 19.7	1.64 (0.103)
	No	76 (38.0)	59.5 ± 11.4		95.1 ± 13.8		112.0 ± 18.9	
Religion	Yes	116 (58.0)	63.5 ± 12.5	1.76 (0.080)	98.8 ± 13.6	1.31 (0.191)	113.7 ± 19.3	–0.92 (0.361)
	No	84 (42.0)	60.4 ± 11.7		96.3 ± 12.3		116.3 ± 19.7	
Education ^‡^	College ^a^	77 (38.5)	62.3 ± 13.5	4.13 (0.018) c > a, b ^‡^	98.1 ± 13.8	2.32 (0.101)	112.9 ± 19.2	0.77 (0.465)
	University ^b^	109 (54.5)	61.1 ± 11.1		96.7 ± 12.6		116.4 ± 19.7	
	Graduate School ^c^	14 (7.0)	70.9 ± 10.1		104.6 ± 11.8		113.2 ± 19.9	
Cohabitant	Yes	163 (81.5)	62.1 ± 12.1	–0.38 (0.706)	97.7 ± 13.0	–0.17 (0.865)	115.0 ± 20.1	0.22 (0.829)
	No	37 (18.5)	62.9 ± 13.0		98.1 ± 13.9		114.2 ± 16.7	
Position	Staff nurse	155 (77.5)	61.9 ± 12.7	3.09 (0.028)	97.5 ± 13.5	2.30 (0.079)	113.7 ± 20.3	1.38 (0.252)
	Charge nurse	15 (7.5)	61.1 ± 12.3		93.5 ± 11.0		115.7 ± 16.1	
	Head nurse	24 (12.0)	61.2 ± 5.9		99.0 ± 11.0		118.1 ± 15.6	
	Chief officer	6 (3.0)	76.8 ± 12.8		109.7 ± 13.1		128.3 ± 17.3	
Nursing experience ^†,‡^		13.4 ± 7.8						
	< 5 years ^a^	29 (14.5)	58.2 ± 13.4	2.42 (0.091)	95.4 ± 13.3	5.19 (0.006) b < c	112.9 ± 16.2	0.25 (0.778)
	5–10 years ^b^	44 (22.0)	61.2 ± 11.3		93.1 ± 12.9		114.1 ± 20.4	
	≥10 years ^c^	127 (63.5)	63.5 ± 12.1		99.9 ± 12.7		115.5 ± 19.9	
Years in LTC hospital		6.5 ± 4.2						
	< 5 years	79 (39.5)	61.5 ± 12.6	0.21 (0.813)	97.1 ± 13.1	0.42 (0.660)	113.0 ± 19.9	0.61 (0.546)
	5–10 years	75 (37.5)	62.6 ± 11.3		97.5 ± 13.6		115.8 ± 18.4	
	≥10 years	46 (23.0)	62.8 ± 13.3		99.3 ± 13.5		116.5 ± 20.6	
COVID-19 Education ^†^	Yes	184 (92.0)	62.6 ± 12.1	1.61 (0.109)	98.4 ± 12.8	2.41 (0.017)	114.8 ± 19.4	0.03 (0.978)
	No	16 (8.0)	57.5 ± 13.4		90.3 ± 14.7		114.7 ± 21.5	
Satisfaction COVID-19 education ^†^	Yes	181 (97.8)	62.7 ± 12.1	1.11 (0.270)	98.7 ± 12.6	2.50 (0.013)	114.6 ± 19.4	–0.20 (0.845)
	No	4 (2.2)	56.0 ± 6.2		82.8 ± 11.4		116.5 ± 27.9	
COVID-19 nursing readiness	Yes	188 (94.0)	62.8 ± 12.0	2.63 (0.009)	98.7 ± 12.4	4.26 (<0.001)	114.1 ± 19.1	–2.12 (0.036)
	No	12 (6.0)	53.3 ± 13.2		82.8 ± 14.8		126.3 ± 22.1	

SD = standard deviation; ^†^ Mann–Whitney and Kruskal–Wallis method. ^‡^ Post hoc analysis using Scheffé and Dunnett T3 methods.

**Table 2 ijerph-18-10327-t002:** Resilience, nursing professionalism, and job stress of participants (*n* = 200).

Variables	Number of items	Total Mean ± SD	Total Range	Item Mean ± SD	Item Range
Resilience	25	62.2 ± 12.2	25–100	2.49 ± 0.49	1–4
Nursing professionalism	29	97.8 ± 13.1	29–145	3.37 ± 0.45	1–5
Job stress	32	114.8 ± 19.5	32–175	3.59 ± 0.61	1–5

SD = standard deviation

**Table 3 ijerph-18-10327-t003:** Correlation between the study variables (*n* = 200).

Variable	Resilience	Professionalism	Job Stress
	r (*p*)		
Resilience	1		
Professionalism	0.42 (<0.001)	1	
Job stress	−0.016 (0.024)	−0.21 (0.003)	1

**Table 4 ijerph-18-10327-t004:** Mediating effect of professionalism on the relationship between the study variables (*n* = 200).

Variables	B	β (*p*)	t	Adj.R^2^	F	*p*
1. Resilience → Professionalism	0.45	0.42 (<0.001)	6.45	0.17	41.6	<0.001
2. Resilience → Job Stress	−0.25	−0.16 (0.024)	−2.27	0.02	5.15	0.024
3. Resilience → Job Stress	−0.14	−0.09 (0.251)	−1.16	0.04	5.13	0.007
Professionalism → Job Stress	−0.25	−0.17 (0.026)	−2.24			
Sobel test: Z = −2.12, *p* < 0.05						

## Data Availability

All data generated or analyzed during this study are included in this published article.

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
