# Peer review of "Effects of the Resilience of Nurses in Long-Term Care Hospitals during on Job Stress COVID-19 Pandemic: Mediating Effects of Nursing Professionalism"

_ijerph, 2021, doi:10.3390/ijerph181910327_

Round 1

Reviewer 1 Report

This study investigates the mediation of professionalism between the relationship between resilience and job stress levels. It is necessary to consider the following points:

First, it is necessary to consider the causal relationship between professionalism and resilience as a moderating variable. Is there any theoretical possibility that professionalism affects resilience?

Logically, can resilience affect professionalism? Since professionalism is a competency acquired through long-term work, while resilience has a strong arbitrary character, is there no possibility of professionalism affecting resilience?

Second, since the coefficient of the influence of resilience and professionalism on stress is not large, is not a more important variable being overlooked?

Third, since this study takes into account the COVID-19 situation, it is necessary to think about how this variable can be directly reflected in the study.

Fourth, recent interest in mediated moderation or moderated mediation rather than simple mediation effect is high. This study needs to consider the application of such advanced model.

Reviewer 2 Report

Dear Author
I read your paper entitled "Effects of the Resilience of nurses in long-term care hospitals during on Job stress COVID-19 pandemic: Mediating Effects of Nursing professionalism". 
The topic is fascinating. Here, I reported my suggestions to improve the quality of the manuscript. I hope that this does not disappoint you.

Introduction 
This section is too long and must be improved. I suggest reducing the length and providing only details able to create interest to continue the reading. Moreover, in the last paragraph, you reported the results of your study, and you should not describe your results in the introduction section.

Materials and methods
1) I appreciated that you performed the sample size calculation, but the number of included patients, such as excluded patients, should be reported in the results section. 
2) Unfortunately, you did not perform any statistical evaluation about the normal distribution of continuous data. The matter is not a secondary issue. Continuous variables should be tested for normal distribution with the Shapiro Wilk test, and data reported as mean and standard deviation or median and quartile distributions. As a consequence, the consequential statistical test should be adapted according to data distribution (i.e. ANOVA vs Kruskal-Wallis, Pearson vs Kendall and so on).   Indeed you wrote that you performed posthoc analysis: which correction for p-value did you adopt? Bonferroni? I suggest improving this section.

Results 
1) Based on my previous reports on the materials and methods section, the authors should improve this section.
2) In addition, I suggest you report 95% confidence intervals for frequencies, means or medians.
3) Data in tables did not help the reading for a clear comprehension of the results. I found it challenging to interpret the data. This section should be improved.

Discussion
This section seemed to be well-written. However, for a coherent elaboration of a detailed report, the author should first re-elaborate the paper according to my suggestions.

In conclusion, the topic was quite interesting, but criticisms existed. Therefore, the paper should be rewritten, and I encourage the author to clearly described statistical methods adopted, such as report results. So, my final decision was "accepted with major revision".

Author Response

Dear editors and reviewer

Thank you for allowing us to submit a revised draft of our manuscript to IJERPH. We appreciate the time and effort that the editor and the reviewers have dedicated to providing valuable feedback on our manuscript. We are grateful to the reviewers for their insightful comments on the paper. We have been able to incorporate changes to reflect all of the reviewers’ suggestions. We have highlighted the changes within the revised manuscript.

Here is a point-by-point response to the editor and reviewers’ comments and concerns.

Point #1

Introduction 
This section is too long and must be improved. I suggest reducing the length and providing only details able to create interest to continue the reading. Moreover, in the last paragraph, you reported the results of your study, and you should not describe your results in the introduction section.

Response #1

Thank you for your comment. Per your comment for the ‘Introduction,’ we edited this section for clarity thoroughly. Also, we removed the sentences that seem to report the result of this study in the last paragraph. (Page:1-2, Line: 25-70)

Point #2

Materials and methods
1) I appreciated that you performed the sample size calculation, but the number of included patients, such as excluded patients, should be reported in the results section. 

Response #2

We agree with your comment. The content has been moved to the result section.
(Page:4, Line:168-170)

Point #3

Materials and methods
2) Unfortunately, you did not perform any statistical evaluation about the normal distribution of continuous data. The matter is not a secondary issue. Continuous variables should be tested for normal distribution with the Shapiro Wilk test, and data reported as mean and standard deviation or median and quartile distributions. As a consequence, the consequential statistical test should be adapted according to data distribution (i.e. ANOVA vs Kruskal-Wallis, Pearson vs Kendall and so on). Indeed you wrote that you performed posthoc analysis: which correction for p-value did you adopt? Bonferroni? I suggest improving this section.

Response #3

Thanks for pointing out this. We performed statistical evaluation about normal distribution of cont. data (with using the Shapiro-Wilk test). Also, we performed consequential statistical test with nonparametric Mann-Whitney test and Kruskal-Wallis test. We used scheffé and Dunnett T3 methods for the posthoc analysis with correction p-value<.05.

(Page: 3-4, Line: 146-165)

Point #4

Results 
1) Based on my previous reports on the materials and methods section, the authors should improve this section.

Response #4

Per your comment, we edited this section for improvement (Page:4, Line:168-170)

Point #5

Materials and methods
2) In addition, I suggest you report 95% confidence intervals for frequencies, means or medians.

Response #5

Thank you for your comment. However, we believe that we can show relevant results of characteristics by means, frequencies, and standard deviations with the characteristics.

Point #6

Materials and methods

3) Data in tables did not help the reading for a clear comprehension of the results. I found it challenging to interpret the data. This section should be improved.

Response #6

Thank you for your comment. However, it is hard to find the challenging part to interpret the data.

To be honest, we referred to previous studies in the preparation and description of this part.

  • Labrague, L. J., & De los Santos, J. A. A. (2020). COVID19 anxiety among frontline nurses: Predictive role of organisational support, personal resilience and social support. Journal of nursing management, 28(7), 1653-1661.
  • Kim, H., & Kim, K. (2019). Impact of selfefficacy on the selfleadership of nursing preceptors: The mediating effect of job embeddedness. Journal of nursing management, 27(8), 1756-1763.

We earnestly request that you consider the correction of this part once again, and if it is nevertheless necessary for correction, please comment on what part it is.

Point #7

Discussion
This section seemed to be well-written. However, for a coherent elaboration of a detailed report, the author should first re-elaborate the paper according to my suggestions.

Response #7

Thank you for your comment. According to your suggestion, we tried to supplement and revise the manuscripts for a more detailed report.

Thank you again for reviewing our manuscript.

Round 2

Reviewer 1 Report

All of things commented are answered and revised